# Alkaloid Extract of *Ageratina adenophora* Stem as Green Inhibitor for Mild Steel Corrosion in One Molar Sulfuric Acid Solution

**Jamuna Thapa Magar** [1] , **Indra Kumari Budhathoki** [2], **Anil Rajaure** [2], **Hari Bhakta Oli** [1,*] 
and **Deval Prasad Bhattarai** [1,*] 

[1]  Department of Chemistry, Amrit Campus, Tribhuvan University, Kathmandu 44600, Nepal
[2]  Department of Chemistry, Mahendra Multiple Campus, Tribhuvan University, Dang 22400, Nepal
*   Correspondence: hari.oli@ac.tu.edu.np (H.B.O.); deval.bhattarai@ac.tu.edu.np (D.P.B.)

**Abstract:** Green corrosion inhibitors are of great interest due to their exciting and environmentally friendly behavior in mild steel corrosion control during and after the acid cleaning process. Herein, alkaloids were extracted from the stem of *Ageratina adenophora* and were ensured by qualitative chemical tests as well as spectroscopic test methods. The corrosion inhibition efficacy of the alkaloids against mild steel corrosion was evaluated by gravimetric, electrochemical and EIS measurement methods. In addition, the adsorption isotherm, free energy of adsorption and thermodynamic parameters of the process were evaluated. The investigations indicated the most promising inhibition efficacy of the alkaloids for mild steel corrosion. The adsorption isotherm study revealed that the adsorption of inhibitor molecules on the MS interface was manifested by dominant physisorption followed by chemisorption. Free energy and thermodynamic parameters are well suited to endothermic processes.

**Keywords:** *Ageratina adenophora*; alkaloids; adsorption isotherm; polarization; EIS





## 1. Introduction

Corrosion is detrimental and threatening to beneficiation, especially in metallic materials, which are crucial in today's industrialized, mechanized, commercialized, and technologically advanced era [1]. The cost of metallic corrosion in terms of monetary loss is approximately US$900 million in Saudi Arabia, US$26.1 billion in India, US$276 billion in the United States, and US$310 billion in China [2,3]. In the context of Nepal, the cumulative annual cost of corrosion corresponds to about 4.3% of the GDP [4]. Moreover, the indirect impacts of metallic corrosion include public service interruptions, accidents, forced shutdowns, the release of toxic substances, and other issues [5]. Sulfuric acid is the most extensively used chemical in the world, with 200 million tons consumed every year. It is typically used in acid pickling, descaling, the production of various chemicals and fertilizers, and the leaching of metallic ores [6]. Mild steel is a preferred, widely used and reasonably priced metallic material for many construction and structural applications due to its mechanical, thermal, magnetic, and electrochemical properties [7]. However, despite its exposure to the environment, it is more vulnerable and susceptible to corrosion through electrochemical reactions [8].

Minimization or inhibition, through various means such as electro-polishing [9], coatings [10], and applications of inorganic and organic inhibitors [11], may be the only way to prevent catastrophes. However, designing economically feasible, sustainable, accessible, efficient, and ecologically safe materials to counteract corrosion through more dependable methods is currently directing corrosion research toward green inhibitors [12]. In addition to other natural products, plant-derived alkaloids are drawing attention as effective corrosion inhibitors [13–17]. Alkaloids are allelochemical compounds mostly

comprising basic nitrogen, oxygen and sulfur atoms with electron-rich centers, making them strong contenders for green inhibitors [18]. Various plant-based alkaloids, such as N-methylisococlaurine [19], Taxine B [20], Voacangine [21] and Tryptamine [22], have been isolated and used as green inhibitors for metallic corrosion in acid media and found to be very efficient. The alkaloids even showed stability in temperatures of up to 58 °C [7]. Their inhibitory effectiveness, however, was reported to be inferior to those of entire plant extracts [19]. Table 1 summarizes part of the recent (from 2020 to 2022 AD) literature on alkaloids as green corrosion inhibitors.

**Table 1.** Some of the literature on alkaloids as green corrosion inhibitors from 2020 to 2022 A.D.

| Plant/Part | Substrate | Medium | Experimental Details | Results | Ref. |
|---|---|---|---|---|---|
| *Coriaria nepalensis*/Stem | Mild Steel (MS) | 1M $H_2SO_4$ | • Alkaloid: 4-pyrimidinecarboxylic acid as reference <br> • Extraction by maceration and solvent extraction techniques <br> • Temperature, concentration and immersion time effect <br> • Methods: Weight loss measurement, potentiodynamic polarization (PDP), and adsorption isotherm | • FTIR: Alkaloids showed various functional groups, e.g., carbonyl group, hydroxyl group, methyl group, amine group, etc. <br> • UV-vis spectrum: Sharp peak at 340 and 422 nm. <br> • $\Delta G_{ads}$: −28.75 kJ/mol <br> • Findings: 1000 ppm gave 96.4% inhibition efficiency (IE) in the weight loss method and 97.03% IE in the polarization test. | [23] |
| *Cinnamomum zeylanicum*/Bark | MS | 1M $H_2SO_4$ | • Alkaloids: Taxine B, Isotaxine B, 2-Deacetyltaxine as reference <br> • Surfactant-based extraction with sonication <br> • Temperature and concentration effect at 2 h immersion time <br> • Methods: Weight loss, PDP, electrochemical impedance spectroscopy (EIS), Scanning Electron Microscopy (SEM), thermodynamic study, quantum chemical calculations, and molecular dynamics (MD) | • Quantum chemical calculations: Lower $E_{HOMO}$ value of Taxine B, and for all alkaloids the fraction of electron transferred ($\Delta N$) value is <3.6, indicating higher electron donation. <br> • Taxine B had a higher binding energy value and strong adsorption. <br> • $\Delta G_{ads}$: −27.31 and −28.15 kJ/mol. <br> • Findings: 200 ppm gave 83.85, 84.49, and 86.12 % IE in weight loss, PDP, and EIS test, respectively. | [24] |
| *Taxus baccata*/Aerial parts | Carbon steel | 1M HCl | • Temperature (°C): 25 <br> • Concentration: 25, 50, 100, 200, 300 and 400 ppm <br> • Methods: Weight loss, PDP, EIS, thermodynamic study | • $\Delta G_{ads}$: −22.03 kJ/mol <br> • Findings: 400 ppm inhibitor gave 97.82% IE and the inhibitor was mixed-type. | [20] |
| *Rhynchostylis retusa* | MS | 1M $H_2SO_4$ | • Alkaloids: Dendroxine as a reference <br> • Extraction: Cold percolation and solvent extraction methods <br> • Temperature (°C): 25 and 35 <br> • Immersion time: 0.5, 3, 6, 9, and 24 h <br> • Concentration: 200, 400, 600, 800, and 1000 ppm <br> • Methods: Weight loss, PDP, optical imaging | • FTIR: Primary, secondary, and tertiary amine groups in the alkaloids are confirmed. <br> • $\Delta G_{ads}$: −27.33 kJ/mol <br> • Findings: <br> • 1000 ppm gave 87.51 and 93.24% I.E. in weight loss and PDP tests. <br> • Optimum temperature for the inhibitor to work is 35 °C. | [25] |

The *Ageratina adenophora* Spreng. (synonym *Eupatorium adenophorum*) plant is a major conspicuous, invasive, prevalent, and alien species known to be of Mexican origin and belonging to the Asteraceae plant family (Figure 1). The plant possesses high adaptability and reproducibility in the invaded environment by modifying the microbial communities in the soil, which has the potential to affect the soil, plants and biodiversity of the area [26,27]. According to the review of the literature on phytochemical [28], phytotoxicity [29], invasive nature, biological control [30] and pharmacological [31] studies, the *Ageratina* plant has

been a prolific producer of numerous primary and secondary bioactive phytochemicals, making it beneficial for antibacterial, anti-inflammatory, antioxidant, and pharmacological purposes. Additionally, this plant has historically been utilized by traditional medicine practitioners for curing various ailments.

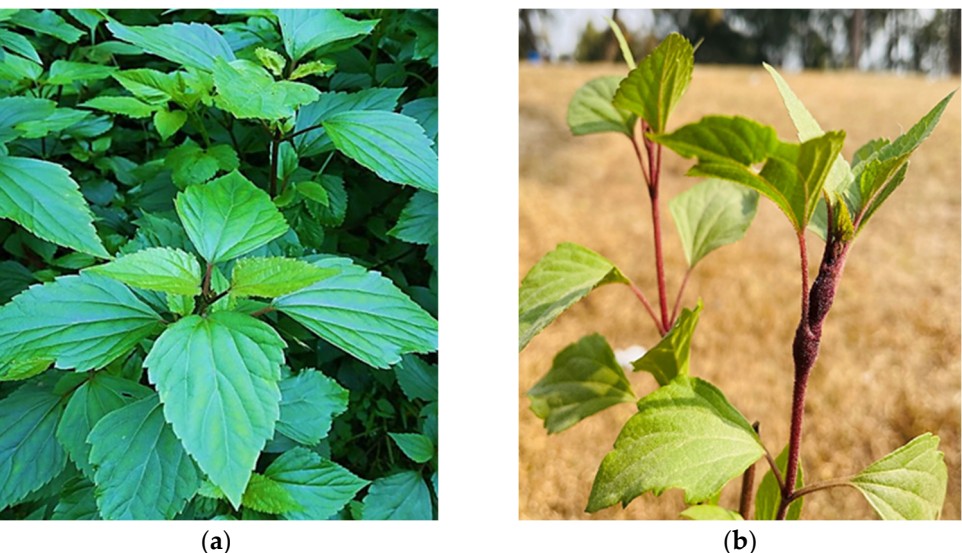

**(a)**     **(b)**

**Figure 1.** (**a**) The morphological structure of the *Ageratina adenophora* plant and (**b**) Small nodules developed on the plant's stem.

A recent study on the phytochemical screening of *Ageratina* leaves reported a higher quantity of alkaloids than other bioactive components such as saponin, tannin and flavonoids [32]. Interestingly, the chemotaxonomic study of this plant's alkaloids also reported the presence of the phosphorus-containing functional groups P-O-P and P-O-C [33], and the phosphorus heteroatoms are considered to have the greatest inhibition efficiency in a series of heteroatoms (oxygen < nitrogen < sulfur < phosphorus) [25,34]. Consequently, the plant is a promising green inhibitor. To date, however, there has been no research on *Ageratina adenophora* as a green inhibitor for mild steel corrosion. The chemical structure of variously reported pyrrolizidine alkaloids such as lycopsamine [35], 3-Hydroxy-2-methyl-butyric acid-retronecinester, and other acetyl derivatives [36] in the Genus *Ageratina* is represented in Figure 2.

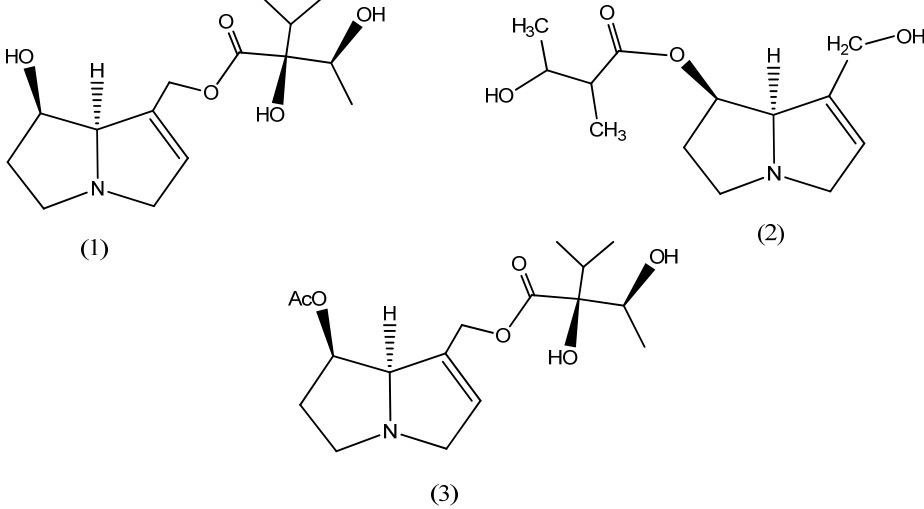

**Figure 2.** Chemical structure of different alkaloids, lycopsamine (**1**), 3-Hydroxy-2-methyl-butyric acid-retronecinester (**2**), and Acetyl-lycopsamine (**3**), in the *Ageratina adenophora* plant.

## 2. Materials and Methods

### 2.1. Chemicals and Instruments

For the extraction and characterization of the alkaloids as well as to prepare the corrosive media, the chemicals and instruments were used as described in the literature [7]. Sulfuric acid (Fischer Scientific, Hampton, NH, USA, 97%, sp. gr. 1.835), oxalic acid (Fischer Scientific, 99%), and sodium hydroxide (Merck life Science, Darmstadt, Germany, 97%) were used to prepare the corrosive media. Dichloromethane (Galaxo Laboratories, Greenford, UK, sp. gr. 1.326), methanol (Thermo Fischer Scientific, Waltham, MA, USA, sp. gr. 0.792), tartaric acid (RANBAXY Lab., Gurgaon, India, 99.0%), and ammonia (Sisco Research Lab., Maharashtra, India, sp. gr. 0.91) were used to extract the alkaloids. All the chemicals were of laboratory grade and were used as received, without any purification. Instruments such as UV (Labotronics, LT-2808), FTIR (Perkin Elmer 10.6.2), a rotary evaporator (IKA 10), and the Digital Vernier Caliper were used as analytical tools. For the electrochemical measurements (i.e., potentiodynamic polarization and electrochemical impedance spectroscopic analyses), the Gamry Framework analyst program version 7.9.0 was used.

### 2.2. Plant Collection and Alkaloid Extraction

The stems of *Ageratina adenophora* (AA) were collected from Ghorahi (Latitude: 28.0583° N, Longitude: 82.4880° E), Dang, Nepal. The shade-dried plant's stems were milled into a powder. One hundred grams of the powder sample was soaked in 750 mL of methanol solution for a week. The alkaloid was extracted from the methanol extract according to the procedure outlined in the literature [23]. The filtrate obtained after the filtration of the mixture was then collected in a beaker and acidified with 5% tartaric acid until the pH reached 3.0. Alkaloids were precipitated in the form of salt. The mixture was then filtered. The residue was taken, and ammonium hydroxide solution was added to maintain pH 10. The alkaline solution was subjected to solvent extraction with an equal volume of dichloromethane (DCM), i.e., alkaline solution, and DCM in (1:1) ratio. The DCM layer contained alkaloids which were collected in a beaker and concentrated using a rotary evaporator. The concentrated solution was then dried using a water bath at 40 °C up to dryness.

### 2.3. Qualitative Tests for As-Prepared Alkaloid

Preliminary chemical analyses such as Mayer's test, Dragendroff's test, and Wagner's test were carried out for the confirmation of alkaloid phytoconstituents. Furthermore, UV-visible spectrophotometry and Fourier Transform Infrared Spectroscopy were performed for the spectrophotometric characterizations of the alkaloid. The UV Spectrophotometer (Labtronics, LT-2802) was used to record the UV-Vis spectrum of alkaloids in the 200–800 nm wavelength range. Similarly, the Perkin–Elmer Spectrum IR version 10.6.2 was used to record spectral data from a 450–4000 $cm^{-1}$ wavenumber range with 4 $cm^{-1}$ resolution, but, before each experiment in spectroscopy, the background correction was performed using isopropanol solvent.

### 2.4. Preparation of MS Specimen, Corrosive Media, and Inhibitor Solution

Mild steel coupons were purchased from the local store in Kathmandu, Nepal. Each coupon was then resurfaced with silicon carbide paper of varying grits (80–1000) and stored in the moisture-free desiccator. The MS coupons were sonicated in ethanol for 10–15 min, dried, and their dimensions were recorded using digital Vernier calipers before each weight loss and electrochemical experiment. The analytical grade of commercial solutions of $H_2SO_4$ (Fisher Scientific, 97%, sp. gr. 1.835) was used to prepare 1M $H_2SO_4$ as corrosive medium through the dilution process. The inhibitor solution was prepared by dissolving one gram of as-prepared alkaloids in a minimum volume followed by dilution up to 1000 mL by using 1M $H_2SO_4$. The resultant solution was 1000 ppm of stock solution of AA. From this stock solution, a series of different concentrations (i.e., 200, 400, 600, and 800 ppm) were prepared by dilution.

### 2.5. Weight Loss Measurement

The methodology described in the literature [7] was adopted for assessing weight loss measurements. The previously abraded, sonicated, dimensioned, and initially weighed MS coupons were submerged in the acid and various inhibitor solution concentrations (200, 400, 600, and 800 ppm) for varying lengths of time (0.5, 1, 3, 6, 9, 18, and 24 h). After immersing the MS coupons for the designated amount of time, each coupon was removed, left to air dry, and then reweighed. Similarly, the temperature effect was investigated by immerging MS coupons in various inhibitor solution concentrations for 1 h at 28, 38, and 48 °C, respectively. The acquired data were used to assess the corrosion kinetics, thermodynamic parameters, and adsorption isotherms.

The following Equation (1) was used to assess the weight loss data to calculate corrosion inhibition efficiency (IE%).

$$\text{Inhibition efficiency (IE\%)} = \frac{w_a - w_p}{w_a} \times 100 \tag{1}$$

where $W_a$ and $W_p$ are the weight loss of MS in acid and different concentrations of inhibitor solutions.

### 2.6. Electrochemical Measurements

The electrochemical measurements were performed by the Gamry Interface 1010B Potentiostat/Galvanostat instrument operated by the software Gamry Framework 7.9.0 analyst version at the APY Laboratory, Central Department of Chemistry, Tribhuvan University, Nepal. A traditional three-electrode electrochemical setup was established by taking MS as the working electrode, platinum (Pt) as the auxiliary electrode, and saturated calomel electrode (SCE) as the reference electrode. The corrosive media and inhibitor solutions were used as the electrolytes in which the pretreated MS samples were submerged for one hour. Afterward, the electrochemical setup was enclosed in a CHI picoamp booster and Faraday cage to prevent electrical interference and frequency noise. Since potentiostatic electrochemical impedance spectroscopy (EIS) is a non-destructive analytical technique, it was carried out before potentiodynamic polarization (PDP) for all one-hour immersed MS samples.

An open circuit delay was carried out for 99 s before each EIS analysis. The EIS curve was then recorded in the frequency range of 100 Hz to 0.01 Hz by applying an AC voltage of 10 mV rms at a scan rate of five points per decade, and the inhibition efficiency was computed using the following Equation (2):

$$\text{Inhibition efficiency (IE)} = \left(1 - \frac{R_{ct}}{R_{ct(inh)}}\right) \times 100 \tag{2}$$

where $R_{ct(inh)}$ ($\Omega$ cm$^2$) and $R_{ct}$ ($\Omega$ cm$^2$) represent the charge transfer resistance of the MS samples in the presence and absence of inhibitor solution, respectively.

The polarization curves were recorded by scanning in the potential range from $-0.8$ to $-0.2$ V at a scan rate of 1 mV/s, and the inhibition efficiency (IE) was computed from the following Equation (3):

$$\text{Inhibition efficiency (IE)} = \left(1 - \frac{I_{corr\ (inh)}}{I_{corr}}\right) \times 100 \tag{3}$$

where $I_{corr(inh)}$ (A cm$^{-2}$) and $I_{corr}$ (A cm$^{-2}$) are the corrosion current density of the MS samples in the presence and absence of inhibitor solution, respectively.

## 3. Results

### 3.1. Alkaloid Characterization

The qualitative chemical tests of the as-prepared alkaloids obtained from the *Ageratina adenophora* (AA) stem gave a colorful precipitate with different chemical reagents [23,25] and confirmed the presence of alkaloids, as displayed in Table 2.

**Table 2.** Confirmatory test for the alkaloids extracted from the AA stem.

| Modes | Mayer's Test | Dragendroff's Test | Wagner's Test |
|---|---|---|---|
| Observation | | | |
| Precipitate Inference | Orange Presence | Orange-red Presence | Reddish brown Presence |

The colored precipitate is due to the formation of complexes between the several alkaloid molecules with the respective chemical reagents. The chemical reactions involved in the phytochemical screening of alkaloids can be simplified and illustrated by considering the lycopsamine compound as a reference alkaloid, as seen in Figure 3. The precipitate formation is due to the formation of potassium salt of alkaloid [23,25,37], and the color development is due to charge transfer/electron transfer reaction within the salt [38,39].

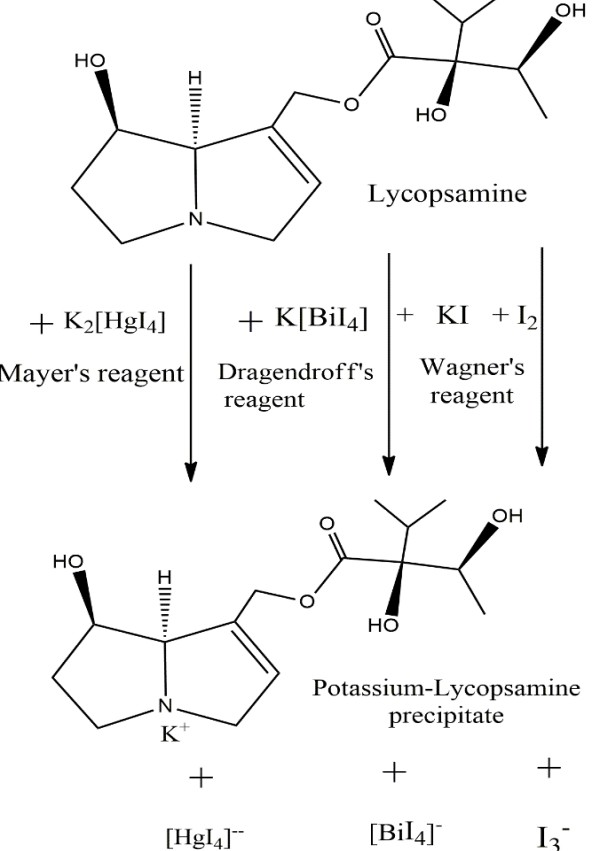

**Figure 3.** Lycopsamine as reference alkaloid participating in the different qualitative chemical reactions.

The interaction of the as-prepared alkaloid with UV or visible light produced spectra with a broad peak at 305 nm is shown in Figure 4. This is manifested by the *n*-π* electronic transition suggesting the presence of a lone pair of electrons or unsaturation in the molecules [23,40,41]. If lycopsamine is used as a reference, then this peak signifies the presence of a polyphenolic group and an aromatic ring with N as a heteroatom.

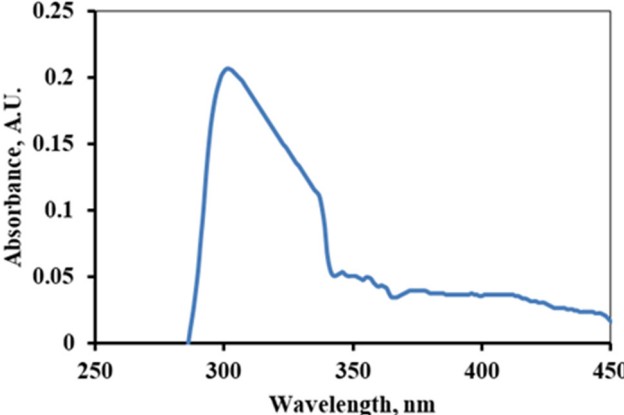

**Figure 4.** UV-Spectra of the alkaloids extracted from *Ageratina adenophora* stem.

The Fourier Transform Infrared Spectroscopic measurement was performed on the polished MS specimen, pure alkaloid or inhibitor, as well as on the MS surface after 1 h immersion in a 600 ppm inhibitor solution, as shown in Figure 5. The broad peak of pure alkaloid at 3385 cm$^{-1}$, in the range of 3469–3272 cm$^{-1}$, is related to the O-H and N-H stretching vibration of the hydroxyl and primary amine functional groups. Another sharp peak at 2925 and 2859 cm$^{-1}$ are due to C-H stretching vibrations by the aliphatic asymmetric hydrocarbon group. The sharp peaks at 1717 and 1663 cm$^{-1}$ are related to the stretching vibrations of the asymmetric and symmetric carbonyl groups (C=O). Similarly, the absorption peak at 1459 cm$^{-1}$ is related to the O-H bending vibration, and a sharp peak at 1374 cm$^{-1}$ is related to the aldehydic C-H bending vibrations. In the 1240–1064 cm$^{-1}$ range, multiple peaks are observed among which an elongated peak at 1064 cm$^{-1}$ is due to the C-N stretching vibrations [40]. Furthermore, a small sharp peak at 977 cm$^{-1}$ could be due to the P-O-P or C-N-C group [42], but a broad peak around 700 cm$^{-1}$ and a sharp peak in the 1050–970 cm$^{-1}$ is absent, indicating the absence of the P-O-P group but the presence of the C-N-C group [40].

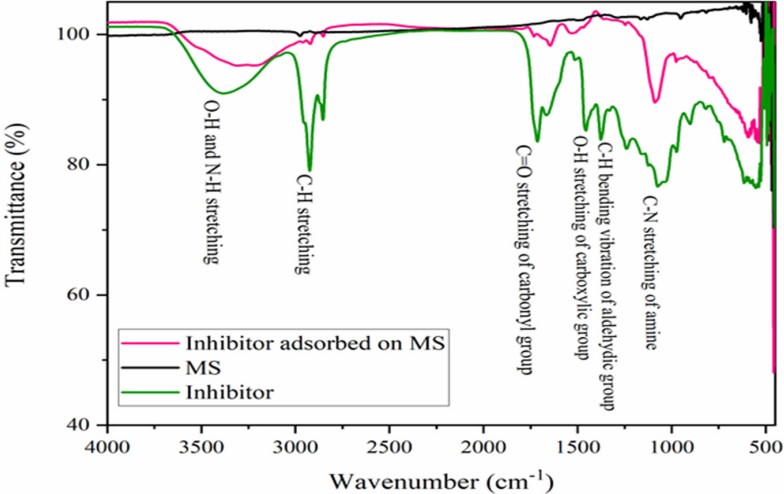

**Figure 5.** FTIR spectra in the 4000–450 cm$^{-1}$ frequency range of MS, inhibitor molecules, and MS immersed in a 600 ppm inhibitor solution for 1 h immersion.

FTIR spectra of inhibitor adsorbed on the MS substrate demonstrates a broad peak around 3397–3288 cm$^{-1}$, indicating the adsorption of the O-H and N-H groups of the inhibitor molecules on the MS. Similarly, the adsorption of the C=O and C-N groups was observed by the adsorption band at 1645 and 1084 cm$^{-1}$. However, there are no significant peaks on the polished MS surface. This indicates that a thin film of alkaloids had developed on the MS surface after immersion [17]. The functional groups of organic moieties, particularly O-H, N-H, C=O and C-N, are responsible for drawing vacant d-orbitals of MS to them during the formation of adsorptive thin films.

### 3.2. Weight Loss Measurements
#### 3.2.1. Effect of Inhibitor Concentration and Immersion Time

The corrosion inhibition efficiency of the alkaloid is a concentration-dependent parameter [19,25]. The observation showed the accelerated inhibiting performance of the inhibitor's concentrations in a series of 200 ppm < 400 ppm < 600 ppm < 800 ppm, as depicted in Figure 6a. The highest inhibiting performance was shown by the 800 ppm (96.95% IE) inhibitor solution due to the abundance of alkaloid molecules which almost completely covered the MS surface for a 1 h immersion period during which the lowest inhibition was seen for the 200 ppm (84.93% IE) inhibitor solution. The longest immersion period during which substantial efficiency (above 70% IE) was seen was 18 h, except for the 200 ppm inhibitor solution. However, after 24 h, the inhibitory performance of the 800 ppm inhibitor concentration declined to 77.88%, and the 200 ppm inhibitor concentration to 40.89%. This is because there are fewer inhibitor molecules to cover the MS substrate, allowing the acid's corrosive attack to happen [23].

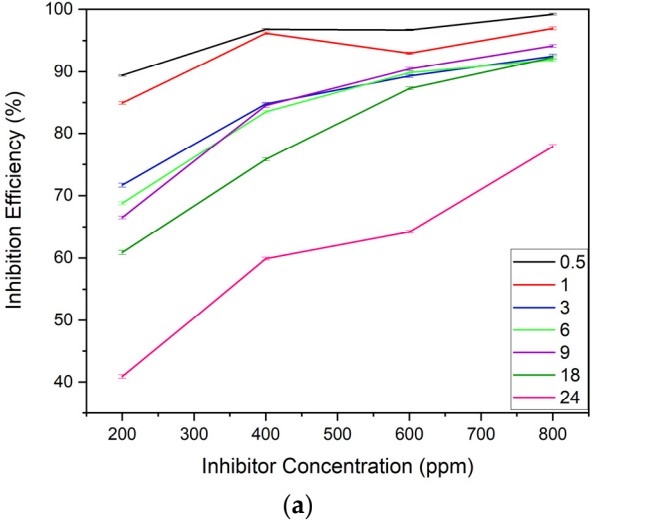

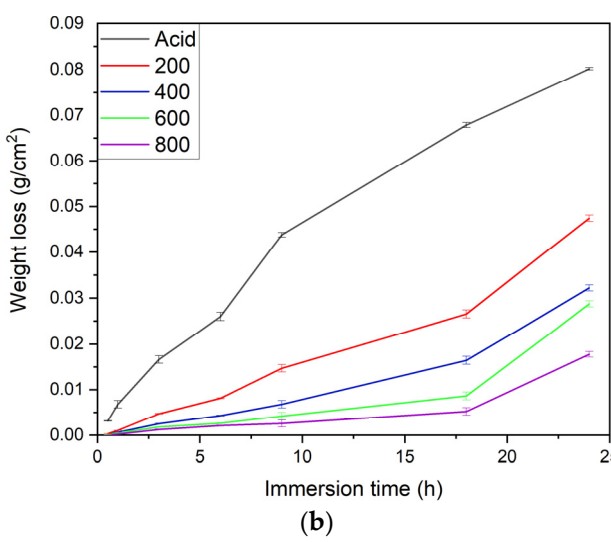

(**a**)                                                                                                             (**b**)

**Figure 6.** Effect of (**a**) concentrations of inhibitor on the I.E. for MS corrosion in 1 M H$_2$SO$_4$, and (**b**) immersion time on the weight loss of MS samples in acid and different inhibitor solutions (*n* = 3 for each assay).

In general, Figure 6b illustrates a noticeable variation in weight loss per unit area of the MS sample exposed to corrosive and inhibitor solution for the prolonged immersion time, indicating an accelerated corrosion rate when inhibition efficiency decelerates. In this experiment, the MS sample dipped in acid solution showed explicitly higher weight loss compared to the corresponding sample in inhibitor solution for all immersion times. The weight loss per unit area curve of the MS substrate was in a sequential order of acid > 200 ppm > 400 ppm > 600 ppm > 800 ppm. When the MS substrate was submerged in an acid solution for 18 h, the weight loss was 0.0678 g/cm$^2$, while the weight loss for MS submerged in an 800 ppm inhibitor solution for the same period was 0.0052 g/cm$^2$. The difference in weight loss caused by the inhibitor's presence is 0.0626 units. This is due to the formation of adsorptive layers, which masked the reactive metal ions from

the corrosive media that reduces the weight loss from the substrate, thus increasing the inhibition efficiency [17,25].

However, when immersion time increased, a sudden decrease in weight loss and inhibition efficiency was observed; this may be due to the following two reasons. The first reason may be the inhibitor molecule's simultaneous desorption from the sample surface triggering a subsequent dissolution in the exposed area [7], and the second one may be the inhibitor molecules' large size, orientation, and interaction, which results in a particular type of defect on the inhibitor layer [25].

### 3.2.2. Effect of Temperature

The effect of temperature on the inhibitor solutions' inhibitory action depends on the molecular vibration, structure, and concentrations of the alkaloids [7]. In this experiment the rise in temperature from 28–48 °C also showed a variation in the weight loss and inhibition efficiency for the MS samples in both acid and inhibitor solutions, as depicted in Figure 7a,b. Here, the sample in inhibitor solution shows implicit weight loss at room temperature 28 °C. At 38 °C, the weight loss of MS dipped separately in acid solution, and in the 200 ppm inhibitor solution this was almost the same. This may be due to the rapid desorption of inhibitor molecules from the substrate. The sample in the 800 ppm inhibitor solution showed minimal weight loss, and other inhibitors (i.e., 400 and 600 ppm) showed relatively moderate weight loss. As shown in Figure 7a, more pronounced nonlinearity in the efficiency of 200 ppm and less in 800 ppm of inhibitor were observed, but, from 38–48 °C, linearity was observed for samples in the 400, 600, and 800 ppm inhibitor concentrations. Furthermore, the observation showed a 20–50% decrease in inhibition efficiency from 28–38 °C; afterward, only a 1–8% decrease was seen. This indicates that the suitable temperatures for the formation of the inhibitor layer are 28 and 48°C.

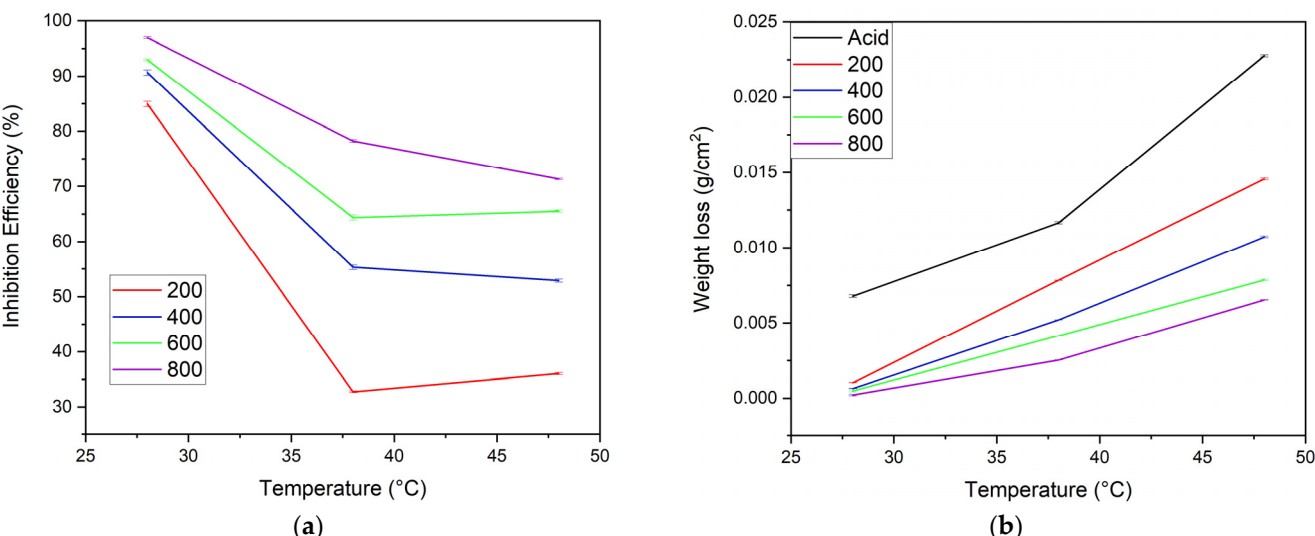

**Figure 7.** Effect of temperature (**a**) on the IE of different concentrations of inhibitor and (**b**) on the weight loss of MS samples in acid and different inhibitor solutions (*n* = 3 for each assay).

The inhibition performance of the alkaloid is primarily due to physisorption followed by chemisorption which in turn is related to the molecular vibrations [25], so temperature-induced physical desorption from the substrate is feasible due to increased molecular vibrations at high temperatures. Not limited to this, at higher temperatures the decomposition of alkaloid molecules may also occur [43].

### 3.2.3. Adsorption Isotherm

Adsorption isotherm models are important graphical representations for understanding the fundamentals of the adsorptive properties of the organic compounds on the MS

surface [43,44]. The inhibitor solution was made by dissolving alkaloid in an acid solution; it has been suggested by Karki et al. [17] that there always exists an equal possibility for the simultaneous adsorption and replacement between the water and inhibitor molecules on the MS surface. The weight loss measurement method was used to evaluate the adsorption isotherms where the fraction of surface coverage ($\theta$) and inhibitor concentration were used. In this experiment, the molecular weight of the alkaloid Lycopsamine was used as a reference to calculate the molar concentration of the inhibitor solution. The Langmuir adsorption isotherm Equation (4) was used for the plot of $\frac{C_{inh}}{\theta}$ against $C_{inh}$ (mol/L) as shown in Figure 8a. A straight line with a regression coefficient ($R^2$) value close to unity (0.999) was obtained. This suggests a process of formation of monolayers before multilayers on the equivalent adsorption sites [17,45,46].

$$\frac{C_{inh}}{\theta} = \frac{1}{K_{ads}} + C_{inh} \tag{4}$$

where $C_{inh}$ represents the inhibitor's concentrations (mol $L^{-1}$) and $\theta$ represents surface coverage. The $K_{ads}$ represents the Langmuir adsorption equilibrium constant from which the Gibbs free energy of the adsorption ($\Delta G_{ads}$) was determined using Equation (5).

$$\Delta G_{ads} = -RTln(55.55 \times K_{ads}) \tag{5}$$

where R is the ideal gas constant (J $mol^{-1}$ $K^{-1}$). The calculated value of adsorption equilibrium and free energy of adsorption is 3333.33 L $mol^{-1}$ and $-30.35$ kJ $mol^{-1}$, respectively, at 301 K. The obtained value of $\Delta G_{ads}$ is less than 40 kJ $mol^{-1}$ and greater than 20 kJ $mol^{-1}$. This indicates a weak electrostatic interaction, accompanied by the chemical interaction of lone pairs of an electron at the electronegative sites of alkaloid molecules with the MS sample [25,47], thus developing a thin protective layer on the MS interface.

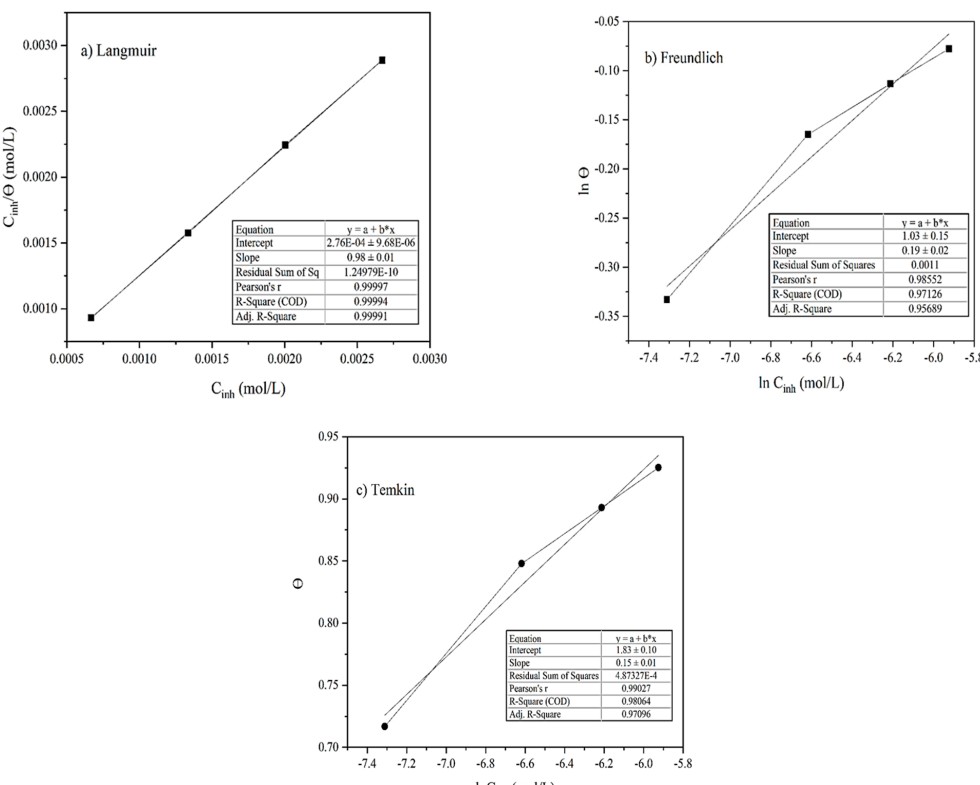

**Figure 8.** Adsorption isotherm plots of (**a**) Langmuir, (**b**) Freundlich, and (**c**) Temkin.

Similarly, the corrosion inhibition mechanism, the interaction nature, and the adsorption processes' spontaneity were studied from the Temkin and Freundlich adsorption

isotherm models employing Equations (6) and (7). The fitting of the Freundlich isotherm plot of ln θ versus ln $C_{inh}$ yielded a slope of a straight line with an $R^2$ coefficient value in the merely acceptable range (0.971) and a slope $(1/n)$ equal to 0.185 as illustrated in Figure 8b. As the value of $1/n$ lies between 0 and 1, this verified a spontaneous adsorption process [17].

$$\theta = -\frac{1}{2a}\ln C_{inh} - \frac{1}{2a}\ln K \qquad (6)$$

$$\ln \theta = \ln K + \frac{1}{n}\ln C_{inh} \qquad (7)$$

where '**a**' is an interaction parameter.

The fitting of the Temkin plot of θ against $\ln C_{inh}$ gave a slope of a straight line with an $R^2$ coefficient value near unity (0.981), as shown in Figure 8c. The obtained $R^2$ value, however, was a little lower than the corresponding $R^2$ value observed in the Langmuir model, so the adsorption of inhibitor extracted from the AA onto the MS obeys the Langmuir model. The calculated values of interaction parameters (a) and K from the intercept and slope were found to be negative 3.31 and 181,679.60 indicating the strong attraction and adsorption of the inhibitor molecules on the MS surface [17,48].

### 3.2.4. Corrosion Kinetics

The corrosion kinetic and thermodynamic activation parameters were elucidated by considering the data obtained from the weight loss measurements at different temperatures. The corrosion rate (CR) was calculated using Equation (8),

$$\text{Corrosion rate (CR)} = \frac{K \times W}{A \times T \times D} \qquad (8)$$

where K is the corrosion rate constant (87,600), W is the mass loss of the MS (g), A is the total surface area of each coupon in (cm$^2$), T is the submerging time (h), and D is the density of MS (g/cm$^3$).

The activation energy of the corrosion reaction was calculated by using the Arrhenius Equation (9).

$$\log(CR) = \log(A) - \frac{E_a}{2.303RT} \qquad (9)$$

where A represents Arrhenius's pre-exponential constant, $E_a$ represents the activation energy, and T is the absolute temperature. From a plot of log (CR) against $1/2.303RT$, a slope of a straight line was obtained from which activation energy was determined. The Arrhenius plot in Figure 9 shows that the energy of activation of the reaction between MS and the acid solution was 48.54 kJ/mol.

This activation energy increases with the addition of inhibitors to 107.26, 114.14, and 113.14, and reaches 139.28 kJ mol$^{-1}$ for 200, 400, 600, and 800 ppm inhibitor solutions, respectively. This increment in activation energy indicates a decrease in the dissolution of MS in acid media. These calculated values lie in between physical and chemical adsorption. Therefore, the adsorption of alkaloids on the MS surface in one molar acid solution is mixed-kind adsorption with dominant chemisorption [17,49].

### 3.2.5. Thermodynamics of Corrosion

The transition state Equation (10) was used to determine the enthalpy and entropy of the system.

$$\log\left(\frac{CR}{T}\right) = \log\left(\frac{R}{hN}\right) + \left(\frac{\Delta S^\circ}{2.303R}\right) - \frac{\Delta H^\circ}{2.303RT} \qquad (10)$$

where h is Plank's constant ($6.6261 \times 10^{-34}$ J s) and N is Avogadro's number ($6.0225 \times 10^{23}$ mol$^{-1}$). The transition state graph is fitted by plotting $\log\left(\frac{CR}{T}\right)$ against $\frac{1}{2.303RT}$. The enthalpy of activation ($\Delta H^\circ$) is measured from the slope of the straight line while the

entropy of activation (ΔS°) is measured from its intercept in the absence and presence of inhibitors, as seen in Figure 10.

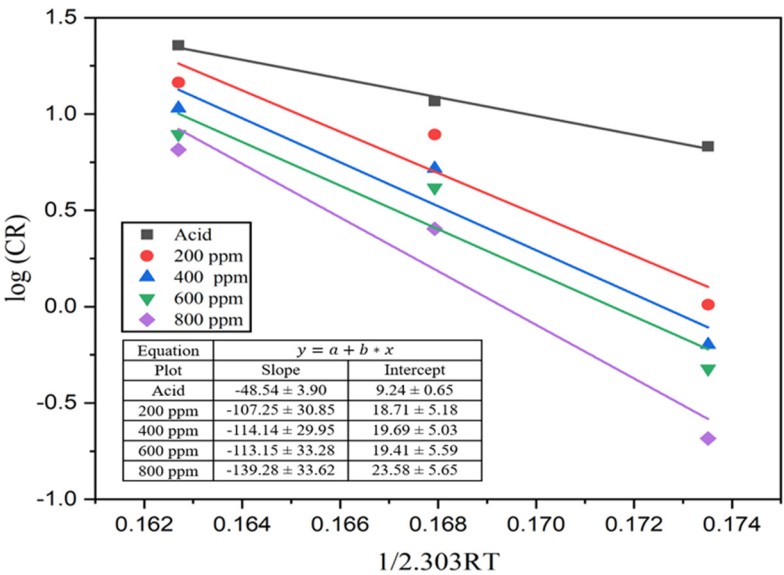

**Figure 9.** Arrhenius plot for the MS sample in 1M $H_2SO_4$ and different concentrations of the inhibitor solutions.

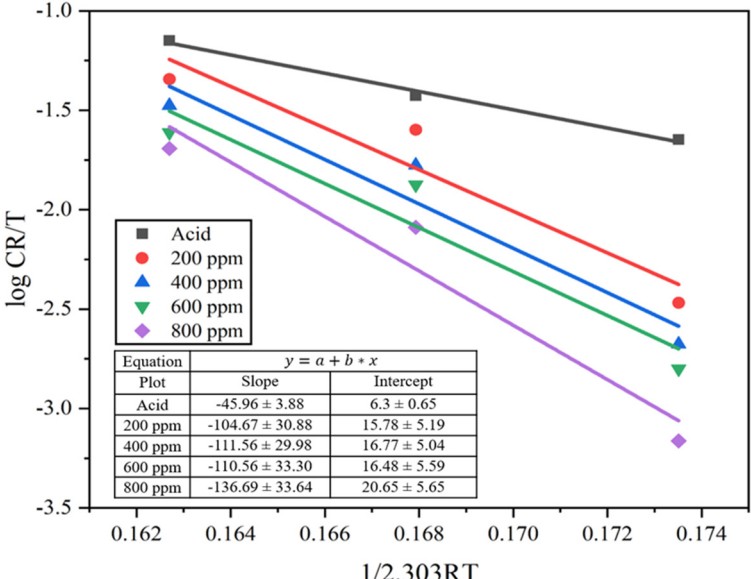

**Figure 10.** Transition state plot for the MS sample in 1M $H_2SO_4$ and different concentrations of the inhibitor solutions.

From the observation, the enthalpy (ΔH°) of the system in the presence of an inhibitor appeared higher than in the absence of an inhibitor. There was a gradual increase in the enthalpy value with an increase in inhibitor solution from 45.96 to 136.69 kJ mol$^{-1}$. Here, the positive value of enthalpy (ΔH°) indicates that the adsorption took place through an endothermic process and reduced the corrosion rate [49]. Similarly, the entropy of the system (ΔS°) is calculated from the intercept of the transition state plot. The entropy increased from negative 76.63 kJ mol$^{-1}$ of acid solution to 104.69, 123.55, 118.08, and 197.96 kJ mol$^{-1}$ for 200, 400, 600, and 800 ppm inhibitor solution. This represents a huge difference in value as it moves from the negative to positive entropy value. This is due to an increase in randomness in the transition state by the formation of activated complexes, i.e., associative mechanisms [17,45] and to the free protons roaming in the solution [7].

This may be due to the replacement of water molecules on the MS surface by the alkaloid molecules during the adsorption process; the process is quasi-substitution [43].

The calculated values of $E_a$, $\Delta H^\circ$, and $\Delta S^\circ$ are presented in Table 3. Here, the $E_a$ value is higher than $\Delta H^\circ$, indicating a cathodic hydrogen evolution reaction leading to a decrease in total reaction volume. The average value of $E_a - \Delta H^\circ$ is 2.58 kJ mol$^{-1}$, which is very close to the value of RT; this obeys the relation $E_a - \Delta H^\circ = RT$. Thus, the adsorption of alkaloids occurs as the physical dominant chemical adsorption [7,45].

**Table 3.** Activation parameters of MS dissolution in 1 M $H_2SO_4$ without and with inhibitor.

| Medium | A (g/cm$^2$) | Ea (kJ/mol) | $\Delta H^\circ$ (kJ/mol) | Ea–$\Delta H^\circ$ | $\Delta S^\circ$ (J/mol K) |
|---|---|---|---|---|---|
| Acid | 32.98 | 48.54 ± 3.90 | 45.96 ± 3.88 | 2.58 | −76.63 |
| 200 ppm | 31.24 | 107.26 ± 30.85 | 104.67 ± 30.88 | 2.59 | 104.69 |
| 400 ppm | 29.21 | 114.14 ± 29.95 | 111.56 ± 29.98 | 2.58 | 123.55 |
| 600 ppm | 30.36 | 113.14 ± 33.28 | 110.57 ± 33.30 | 2.57 | 118.08 |
| 800 ppm | 29.08 | 139.28 ± 33.62 | 136.70 ± 33.64 | 2.58 | 197.96 |

*3.3. Electrochemical Measurements*

The corrosion kinetics and inhibition type of the MS sample in 1 M $H_2SO_4$ solution with and without different concentrations of alkaloid extracted from the AA was investigated by PDP and EIS measurements at 293 K.

3.3.1. Potentiodynamic Polarization Tests (PDP)

The pretreated MS sample with the acid and corresponding inhibitor solutions (200, 400, 600, and 800 ppm) were placed in an electrochemical setup to record the one-hour immersed polarization curves in the potential window of −0.8 to −0.2 V on the Gamry Framework instrument at 293 K. Triplicate measurements were performed for the reproducibility issue. Figure 11 shows the anodic and cathodic Tafel polarization curves of one-hour-immersed MS sample in acid and different concentrations of inhibitor solution, from which electrochemical parameters such as current density, corrosion potential, anodic slope, cathodic slope, and inhibition efficiency are recorded by the extrapolation method and placed in Table 4. The rapid physical adsorption and desorption of the inhibitor molecules on the MS surface after its exposure to acid and inhibitor media showed a decrement in the corrosion current density ($I_{corr}$) [7,17] with the increment in the inhibitor concentrations.

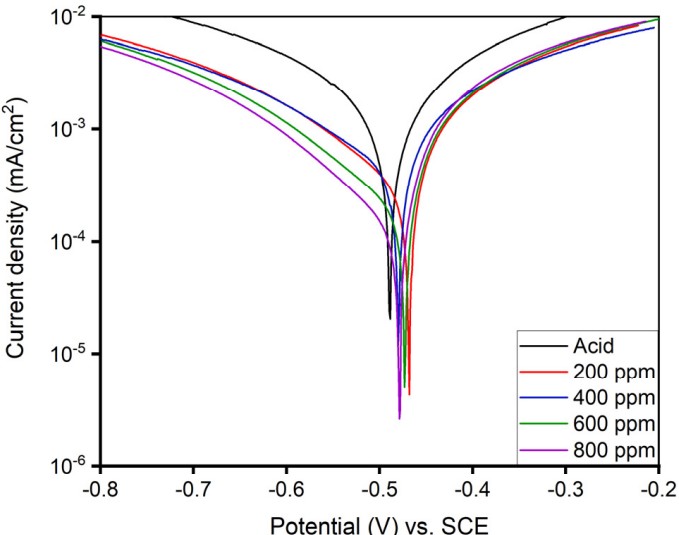

**Figure 11.** PDP curve for as-immersion of MS sample in 1 M $H_2SO_4$ solution with and without different concentrations of alkaloid extracted from AA at 293 K.

**Table 4.** Polarization parameters of 1 h-immersed MS sample in acid solution with and without inhibitor.

| Medium | OCP (V) | Current Density ($\mu A/cm^2$) | Anodic Slope | Cathodic Slope | Efficiency (%) |
|---|---|---|---|---|---|
| Acid | −0.485 | 1.14 | −5.77 ± 0.05 | 7.32 ± 0.09 | – |
| 200 ppm | −0.468 | 0.31 | −6.14 ± 0.03 | 12.19 ± 0.24 | 72.81 |
| 400 ppm | −0.480 | 0.28 | −5.83 ± 0.03 | 18.11 ± 0.41 | 75.44 |
| 600 ppm | −0.472 | 0.18 | −6.67 ± 0.02 | 13.56 ± 0.37 | 84.21 |
| 800 ppm | −0.478 | 0.10 | −7.31 ± 0.04 | 20.17 ± 0.60 | 91.23 |

The Tafel plot of the one-hour-immersed MS sample exhibited a decreasing $I_{corr}$ value with a corrosion potential in the range of −0.468 to −0.490 V as the inhibitor concentration increased. The inhibitory performance of the 800 ppm solution appeared to have the highest efficiency (91.23%) with an $I_{corr}$ value of 0.10 $\mu A/cm^2$. The decreasing $I_{corr}$ value shown in Table 4 is due to the physical adsorption of inhibitor molecules covering the substrate, which reduces electron transfer from the surface, thus preventing potential attack from the corrosive ions, resulting in the increased resistance of MS [7].

3.3.2. Electrochemical Impedance Spectroscopy (EIS)

Electrochemical impedance spectroscopy was performed to estimate the resistance or impedance of the MS at one-hour immersion. Triplicate measurements were performed in order to maintain reproducibility. $R_{ct(inh)}$ represents the charge transfer resistance of the inhibitor molecules on the MS sample. The Nyquist and Bode plots obtained from the EIS study are shown in Figure 12a–c, and the equivalent circuit model is shown in Figure 12d. In the circuit model, Rs represents the solution resistance, Rct represents charge transfer resistance at the metal/solution interface, and CPE is a constant phase element representing the double layer capacitance of the metal/solution interface. The Rct value of MS in acid solution is 6.59 $\Omega$ cm$^{-2}$. The real corrosion system comprises non-homogeneity due to the presence of surface roughness, the adsorption of inhibitors and the formation of a porous layer on the electrode surface. The CPE in the model circuit compensates for this non-homogeneity [50]. The impedance function is represented by the Equation (11) [48]:

$$ZCPE = \frac{1}{Q(j\omega)^n} \tag{11}$$

where Q represents the magnitude of the CPE, j is the imaginary number and $\omega$ is angular frequency, and *n* is the CPE exponent that lies between −1 to +1; this could reflect the non-homogeneity [48,49].

The observation showed an increment in the impedance value with an increase in the concentration of the inhibitor molecules as in Table 5. The maximum inhibitory performance (92.53%) was observed for MS in the 800 ppm solution, while the minimum inhibitory performance was observed for the MS sample in the 200 ppm inhibitor solution. This is due to the development of the thin protecting layer on the MS interface which prevents or eliminates the potential penetration of the corrosive ions or molecules by increasing the interfacial resistance/impedance of the MS sample.

**Table 5.** Electrochemical impedance parameters derived from the equivalent circuit using Z-view software.

| Medium | Rs ($\Omega$ cm$^2$) | CPE ($\mu\Omega$ S$^n$ cm$^{-2}$) | *n* | Rct(inh) ($\Omega$ cm$^2$) | %IE |
|---|---|---|---|---|---|
| 200 ppm | 17.31 | 338.04 | 0.888 | 15.71 | 58.05 |
| 400 ppm | 17.42 | 197.07 | 0.892 | 21.7 | 69.63 |
| 600 ppm | 16.92 | 184.79 | 0.858 | 31.45 | 79.05 |
| 800 ppm | 17.51 | 102.4 | 0.885 | 88.25 | 92.53 |

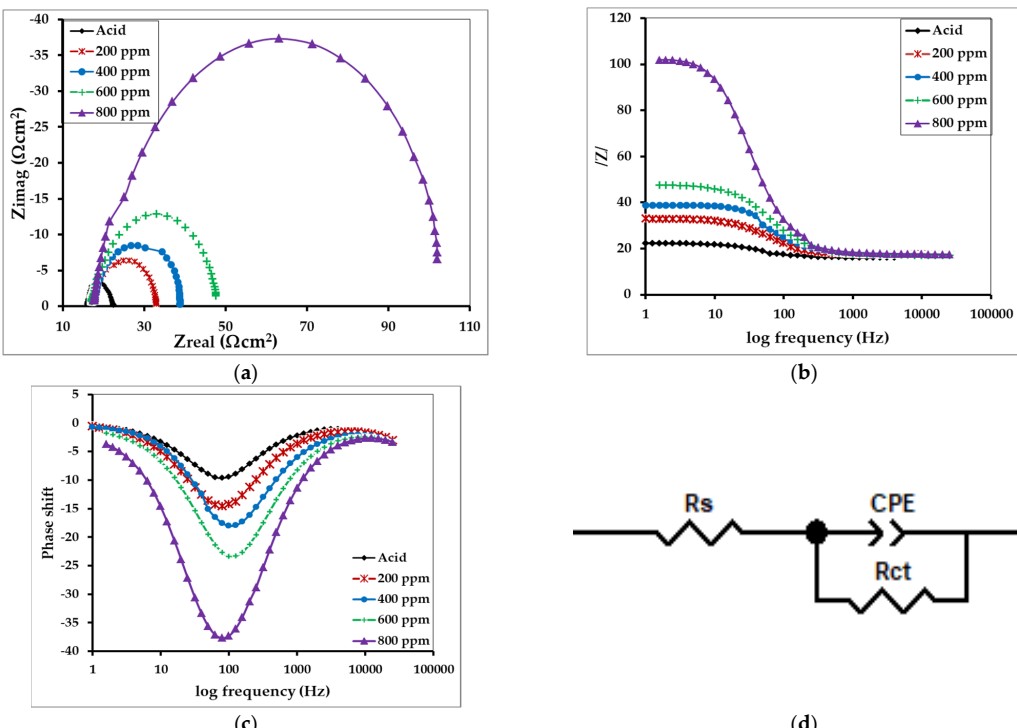

**Figure 12.** Representing (**a**) Nyquist (Cole-Cole) plot, (**b**) Bode plots, and (**c**) Bode plots of phase shift versus log frequency for the MS in 1 M $H_2SO_4$ without and with inhibitors of different concentrations, and (**d**) the equivalent circuit model used to fit the impedance spectra.

It was found that the corrosion of MS in acid medium is controlled by the charge transfer process without changing the mechanism, as indicated by a single capacitive loop (Figure 12a) in the Nyquist plot and only one time constant in the Bode plot [51]. The gradual increase in capacitive loop diameter in the Nyquist plot upon increasing the concentration of inhibitor represents an increment in charge transfer resistance due to more substantial surface coverage by the inhibitor molecules [52]. The increase in the phase angle in the Bode-phase plot also confirms the inhibitive behavior of alkaloids [20,53].

The adsorption isotherms were also fitted from the data obtained by the PDP and EIS measurement methods. The best fitted data were for the Langmuir adsorption isotherm. The $R^2$ values for both cases are nearly equal to unity, indicating that the adsorption followed the Langmuir adsorption isotherm. This finding is similar to the isotherm fitted from weight loss data. Figure 13a,b presents the Langmuir adsorption isotherm fitted from the PDP and EIS data, respectively.

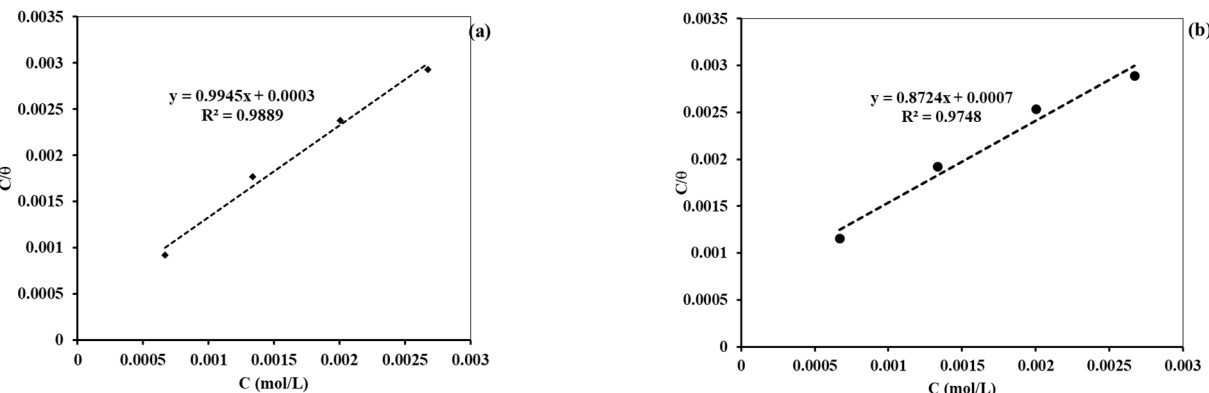

**Figure 13.** Diagram showing the Langmuir adsorption isotherm from (**a**) PDP data and (**b**) EIS data.

## 4. Discussion and Inhibition Mechanism

The maximum efficiency observed in the polarization experiment is, however, smaller compared to that of the EIS tests, which in turn is smaller compared that of the weight loss measurements. The handling and systematic errors may have contributed to differing inhibition efficiencies between the weight loss and PDP tests [25]. Since EIS is considered a more sensitive and accurate electrochemical technique than PDP [54], the efficiency of inhibitors obtained from this method is acceptable.

Corrosion inhibition by organic molecules is described by their adsorption on the MS surface. The presence of heteroelements and the polar functional groups in the organic moieties are responsible for the adsorption process [7,55]. Initially, the protonated alkaloid molecules interact with the negatively charged ions being adsorbed on the MS surface. Then, after the successful release of the proton, the alkaloid molecules are adsorbed on the MS surface, forming a protective layer [56]. This mechanism is supported by thermodynamic values and electrochemical measurements [23,57]. The detailed inhibition mechanism is displayed in Figure 14 below.

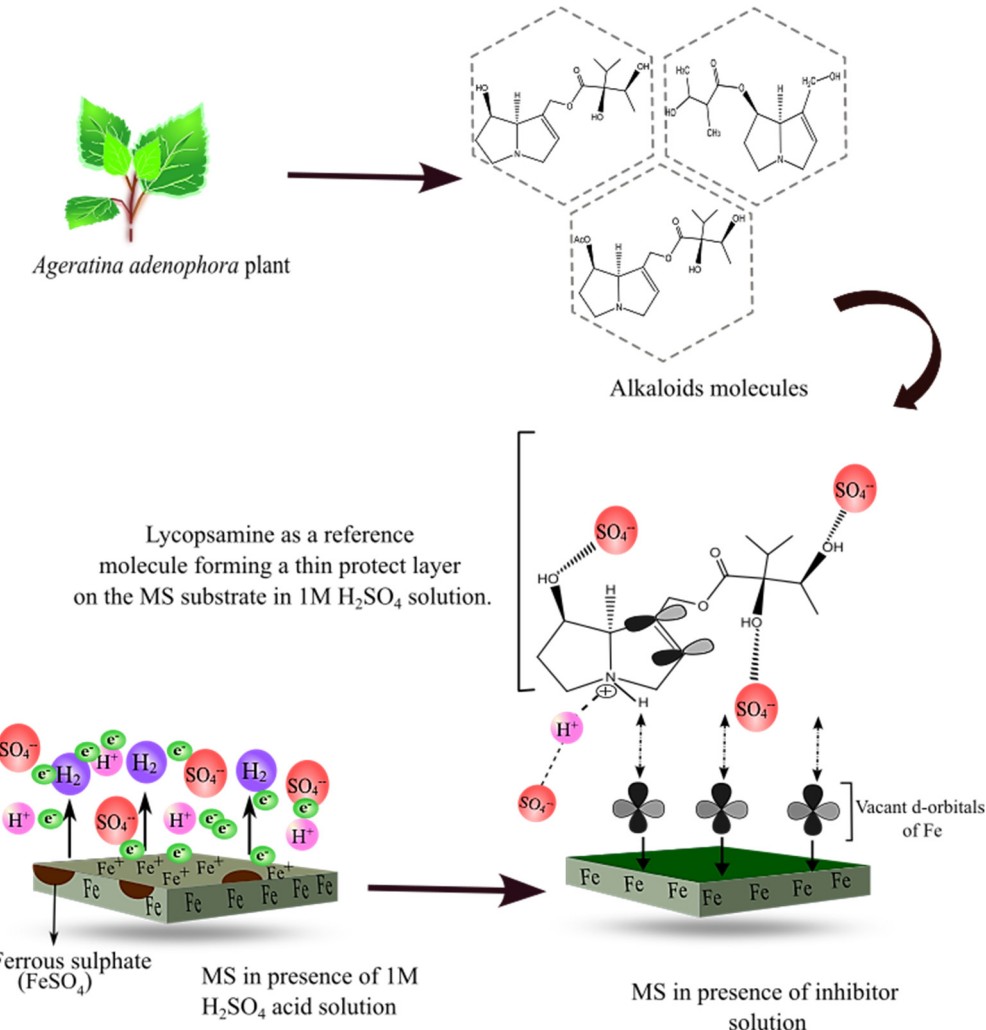

**Figure 14.** Schematic diagram showing the mechanism of corrosion inhibition by alkaloid molecules in reference to lycopsamine molecule.

## 5. Conclusions

An alkaloid as a green inhibitor has been extracted from the stem of the *Ageratina adenophora* plant through the cold percolation method and solvent extraction technique. The chemical and spectroscopic characterizations confirmed the presence of the as-prepared

alkaloid molecules. The maximum corrosion inhibition efficiency (%) of 91.23, 92.53, and 96.95 was exhibited by the 800 ppm inhibitor solution in the polarization, EIS, and weight loss measurements, respectively, at 293 and 301 K. Based on the calculated values of $K_{ads}$ and $\Delta G_{ads}$, the adsorption nature of the inhibitor molecule on the MS interface strongly obeys the Langmuir adsorption isotherm model. Furthermore, the adsorption is spontaneous due to the strong interaction between the inhibitor molecules, as suggested by the Freundlich and Temkin models. It can thus be inferred that the *Ageratina adenophora* alkaloid is a prospective green inhibitor against mild steel corrosion. However, more studies are still needed to identify and determine major alkaloids and their individual effects on inhibitory performance.

**Author Contributions:** Conceptualization, H.B.O. and D.P.B.; methodology, H.B.O.; software, H.B.O.; validation, J.T.M., H.B.O. and D.P.B.; formal analysis, H.B.O.; investigation, J.T.M.; resources, D.P.B.; data curation, J.T.M. and H.B.O.; writing—original draft preparation, J.T.M., I.K.B. and A.R.; writing—review and editing, H.B.O. and D.P.B.; visualization, D.P.B.; supervision, H.B.O.; project administration, D.P.B.; funding acquisition, H.B.O. All authors have read and agreed to the published version of the manuscript.

**Funding:** This research received no external funding.

**Institutional Review Board Statement:** Not applicable.

**Informed Consent Statement:** Not applicable.

**Data Availability Statement:** Will be provided based on 'MDPI Research Data Policies'.

**Acknowledgments:** Authors would like to acknowledge the Department of Chemistry, Amrit Campus for laboratory support, and APY laboratory, Central department of chemistry, TU, for electrochemical work station.

**Conflicts of Interest:** The authors declare no conflict of interest.

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
