# Peer review of "Alkaloid Extract of Ageratina adenophora Stem as Green Inhibitor for Mild Steel Corrosion in One Molar Sulfuric Acid Solution"

_2673-3293, doi:10.3390/electrochem4010009_

Round 1
Reviewer 1 Report
The authors report an extract of Ageratina adenophora as a viable inhibitor of steel corrosion. This approach is well described in the literature. However, the good results obtained through a relatively easy approach may be interesting to the readership of Electrochem.
However, prior to its publication, some questions remain to be addressed by the authors:
1. Please, indicate the number of explored samples for each conditions (n) in each assay. In order to provide meaningful information, at least a triplicate should be measured for every condition.
2. Include the deviation of each measurement according in the corresponding figures (6 and 7)
3. There is no need to use 5 decimal units when your error is in the units. Present the data with significant figures.
Author Response
We have addressed each comment in the revised manuscript.

Reviewer 2 Report
Manuscript ID: electrochem-2219618
Title: Alkaloid extract of Ageratina adenophora stem as green inhibitor for Mild steel corrosion in one molar sulphuric acid solution.
This manuscript presents an experimental work focused on green corrosion inhibitors. Some alkaloids were extracted from Ageratina adenophora and its corrosion inhibition efficacy was tested against mild steel corrosion. Gravimetric, electrochemical and EIS measurements were employed to evaluate the corrosion inhibition efficacy of the alkaloids against mild steel corrosion. The results are relevant. This manuscript is suitable for publishing after minor revisions.
Comments:
1.- In the introduction section, on page 1, lines 31-32, the sentence: “Sulfuric acid is the most extensively used chemical in the world and every year 200 million sulfuric acids are consumed.” the mass units should be added, 200 million tons?
2.- In the introduction section, the meaning of abbreviations MS, PDP, IE, EIS should be indicated the first time that appear in the text with the full name and the abbreviation between parenthesis.
3.- In Table 1, the absence of minus sign is missed in ∆Gads for reference [26]. Please revise.
4.- In section “2. Material and Methods” the authors should provide more information about alkaloid extraction (2.2. Plant collection and alkaloid extraction) and the preparation the corrosive media, chemicals and instrument used (2.1. Chemicals and Instruments).
5.- In section “3.2.1. Effect of Inhibitor Concentration and Immersion Time” the following sentence on page 8, lines 218-220: “However, after 24 hours, the inhibitory performance of the 800 ppm inhibitor concentration declined to 77.88%, whereas the 200 ppm inhibitor concentration was virtually undetectable.” However, in Figure 6a, after 24 hours and 200 ppm inhibitor concentration the inhibitory efficiency is around 40%.
6.- In section “3.2.2. Effect of Temperature”, the authors should explain in detail the sentence on page 9, lines 250-251: “The MS in 200 ppm inhibitor has shown a weight loss curve near the weight loss curve of the sample in acid at 38 ºC …”
7.- The experimental procedure to determine the adsorption isotherm should be explained.
8.- The sentence on page 11, line 299: “Where a is an interaction parameter” appears that is not located in the correct place.
9.- In section “3.2.4. Corrosion kinetics” the authors should explain the procedure to calculate CR and its units.
10.- Reference [44] is the reference [15] repeated.
11.- Change “sulphuric” by “sulfuric” in the title of the manuscript.
Author Response
We have addressed all the comments in the revised manuscript.

Reviewer 3 Report
This is an interesting manuscript on the corrosion inhibition of mild steel in 1M H2SO4 with an extract of Ageratina adenophora. The general impression is that the scientific contribution, originality, quality of structure, and clarity of the whole manuscript are at a satisfactory level.
The research is well-planned and executed. However, in order to improve and clarify the whole manuscript, I would like to add some suggestions:
- In the introductory part, it is necessary to emphasize the purpose as a new scientific contribution of this research.
- The results of the electrochemical measurements (PDP and EIS) need to be provided with an appropriate isotherm. Discuss the similarities/differences between the results obtained by the different test methods.
- Generally, it is not stated how many times each experiment was performed for each inhibitor concentration; it is appropriate to state this and data must be provided for statistical analysis of the tests performed and to verify reproducibility. Please add the error limits for the numerical values in tables 3-5.
Author Response

(The authors gave the same response as above.)

Round 2
Reviewer 1 Report
I'm not sure what the authors mean by the complexity of arranging error bars in a graph. Every graphing software has the option to add the corresponding standard deviation and use that data for plotting the error bars.
This information is relevant in order to properly gauge the effect of the inhibitor. Without this information, the graphs lack any statistical significance.
Error bars (with the standard deviation of each point) should be added prior to the acceptance of this manuscript.
Author Response
Reviewers' comments has been addressed.

Reviewer 3 Report
The authors have revised the manuscript well so that it can be accepted in its present form.
Author Response
Thank you very much for the acceptance of the paper.